# Risk Factors for Early and Late Onset Preeclampsia in Reunion Island: Multivariate Analysis of Singleton and Twin Pregnancies. A 20-Year Population-Based Cohort of 2120 Preeclampsia Cases

**Pierre-Yves Robillard** [1,2,*] **, Malik Boukerrou** [2,3] **, Gustaaf Dekker** [4] **, Marco Scioscia** [5] **, Francesco Bonsante** [1,2] **, Brahim Boumahni** [1] **and Silvia Iacobelli** [1,2]

[1] Service de Néonatologie, Centre Hospitalier Universitaire Sud Réunion, BP 350, CEDEX, 97448 Saint-Pierre, La Réunion, France; francesco.bonsante@chu-reunion.fr (F.B.); brahim.boumahni@chu-reunion.fr (B.B.); silvia.iacobelli@chu-reunion.fr (S.I.)

[2] Centre d'Etudes Périnatales Océan Indien (CEPOI), Centre Hospitalier Universitaire Sud Réunion, BP 350, CEDEX, 97448 Saint-Pierre, La Reunion, France; malik.boukerrou@chu-reunion.fr

[3] Service de Gynécologie et Obstétrique, Centre Hospitalier Universitaire Sud Réunion, BP 350, CEDEX, 97448 Saint-Pierre, La Reunion, France

[4] Department of Obstetrics & Gynaecology, Robinson Institute, Lyell McEwin Hospital, University of Adelaide, North Adelaide, SA 5005, Australia; gustaaf.dekker@adelaide.edu.au

[5] Unit of Gynecological Surgery, Department of Obstetrics and Gynecology, Mater Dei Hospital, 70125 Bari, Italy; marcoscioscia@gmail.com

\* Correspondence: robillard.reunion@wanadoo.fr; Tel.: +262-2-62-35-91-49; Fax: +262-2-62-35-92-93

**Abstract:** Objectives: To develop a multivariate model for risk factors specific to early onset preeclampsia (EOP) and late onset preeclampsia (LOP) in our entire population (singleton and twin pregnancies). Material and methods: 20 year-observational population-based historical cohort study (2001–2020). All consecutive births delivered at the Centre Hospitalier Universitaire Hospitalier Sud Reunion's maternity ward. A standardized validated epidemiological perinatal database was used. Results: During the 20-year period, there were 81,834 pregnancies and 83,497 infants born, 1232 dichorionic and 350 monochorionic twin pregnancies. There were 2120 cases of preeclampsia, of which 2001 were preeclamptic singleton pregnancies and 119 twin pregnancies (incidence 7.5% in twin pregnancies vs. 2.5% singletons, OR 3.0, $p < 0.001$). Independent risk factors for EOP and LOP in a multivariate model (controlling for the two major confounders: maternal ages—both risks for EOP and LOP, and maternal pre-pregnancy BMI—specific risk factor for LOP) were: history of preeclampsia (adjusted OR (aOR) 11.7 for EOP, 7.8 for LOP, $p < 0.0001$), chronic hypertension (aOR 7.3 for EOP, 3.9 for LOP, $p < 0.0001$), history of perinatal death (aOR 2.2 for EOP, $p < 0.0001$ and 1.48 for LOP, $p = 0.007$), primipaternity (aOR 3.0 for EOP and 3.6 for LOP, $p = 0.001$), dizygotic twin pregnancies (aOR 3.7 for EOP, $p < 0.0001$ and 2.1 for LOP, $p = 0.003$), monozygotic twin pregnancies (aOR 3.98 for EOP, $p = 0.003$ and non-significant (NS) for LOP), ovulation induction (aOR 5.6 for EOP, $p = 0.004$ and NS for LOP), and in vitro fertilization (aOR 2.8 for EOP, $p = 0.05$ and NS for LOP). Specific to LOP and NS for EOP: renal diseases (aOR for LOP 2.9, $p = 0.007$) and gestational diabetes mellitus (aOR 1.2, $p = 0.04$). Conclusions: Maternal ages over 35 years, chronic hypertension, history of preeclampsia, ovulation induction, in vitro fertilizations, history of perinatal deaths and twin pregnancy (in our experience, especially mono zygotic twin pregnancies) are significant risk factors for EOP. New paternity is an independent factor for both EOP and LOP.

**Keywords:** early onset preeclampsia; late onset preeclampsia; preeclampsia; epidemiology; twin pregnancies; aspirin; gestational weight gain; preeclampsia; epidemiology; monochorionic twins; dichorionic twins

## 1. Introduction

In Reunion island we have a very high incidence of early onset preeclampsia EOP (31% of our preeclampsia cases, PE) [1–3], which is surprising as compared with international literature [1], where EOP cases has rather proportion of 10%. Over the last five years we published six epidemiological studies analyzing risk factors for EOP and LOP (five on singleton pregnancies [4–8] and the last one analyzing twin pregnancies [9]). The present study wishes to make a global synthesis of all these studies.

The first study [4] revealed that EOP women were older than LOP ones; while primigravidas and primiparas, typically younger than multiparous women, were more prone to develop LOP, findings also confirmed by a study we shared with the University Maternity of Antananarivo, Madagascar [10]. The second study [5] concluded that chronic hypertension and history of preeclampsia were the strongest risk factors for EOP. However, rising pre-pregnancy BMI was only associated with LOP (and very poorly with EOP), and in controlling for maternal ages and booking/pre-pregnancy BMI, diabetes was not an independent risk factor neither for EOP nor for LOP. The third study [6] tested whether it was not the international cut-off of 34 weeks between EOP and LOP that could explain our previous results. We tested then different possible cut-offs (CO) at the 30th, 32nd, 34th, and 37th week of gestation. The major results being similar, the international cutoff of the 34th week of gestation is then appropriate. The fourth study [7] added to the major risks for EOP thrombophilia, induction of ovulation (without IVF) and for LOP women with pre-existing renal disease. In the fifth study [8], we tested what could be the remaining risk factors in a population where major risk factors for EOP (chronic hypertension, history of preeclampsia mainly) and other eight risk factors (multiple pregnancies, pre-existing diabetes mellitus, chronic hypertension, history of previous preeclampsia, 'thrombophilia', renal or thyroid disease, and smoking) were excluded. This population comprised 72% of our female reproductive community, but they still comprised also 63% of all preeclampsia cases [8]. In this selected population, we confirmed that overweight and different classes of obesities were linearly and increasingly linked with only LOP, and disconnected with EOP. New paternity was also an independent factor for both EOP and LOP in multiparas [8]. Finally, our sixth study [9] explored the association between EOP and LOP and twin pregnancies: 1232 dichorionic (DC) and 350 monochorionic (MC) twin pregnancies. There were 2120 preeclampsia cases of which 2001 PE singleton pregnancies and 119 twin pregnancies (incidence 7.5% in twin pregnancies vs. 2.5% singletons, OR 3.0, $p < 0.001$), this relative risk of PE being is well-known in the literature. However, the EOP proportion of PE cases was similar in singleton and DC twin pregnancies: 32% (respectively 641/2001, 32.0% and 26/84, 31.0%, while it was 2/3 of cases in MC twin pregnancies: 17/26, 65.4%, crude OR 4.0 MZ-EOP vs. singleton-EOP, $p < 0.0001$). The present study develops a multivariate model controlling for the two major confounders: Maternal age (both risks for EOP and LOP) and maternal pre-pregnancy BMI (major risk factor for LOP [5]) with all the detected other risk factors in singleton and in twin pregnancies.

## 2. Material and Methods

From 1 January 2001 to 31 December 2020, the hospital records of all women delivered at the maternity ward of the University South Reunion Island (approximately 4300 births per year) were abstracted in standardized fashion. The study sample was drawn from the hospital perinatal database, which prospectively records data from all mother-infant pairs since 2001, and all normotensive singleton pregnancies included as reference. Information is collected at the time of delivery and at the infant hospital discharge and regularly audited by appropriately trained staff. These epidemiological perinatal databases which contained information on obstetrical risk factors, description of deliveries and neonatal outcomes. For the purpose of this study records have been validated and have been used anonymously. Additionally, during all the prenatal follow-up during pregnancy, and as participants in the French national health care system, all pregnant women in Reunion Island have their prenatal visits, biological and ultrasonographic examinations,

and anthropological characteristics recorded in their maternity booklet before coming at the maternity for delivery. Preeclampsia, gestational hypertension and eclampsia were diagnosed according to the definition issued by the International Society for the Study of Hypertension in Pregnancy (ISSHP): blood pressure ≥140 mmHg systolic or ≥90 mmHg diastolic at ≥20 weeks of gestation and proteinuria ≥300 mg/24 h or protein-to-creatinine ratio >30 mg/mmol or ≥2+ on dipstick testing. This has been revised recently by the ISSHP and American College of Obstetricians and Gynecologists to include cases without proteinuria, but with evidence of relative renal, hepatic, or hematological dysfunction. We followed these to the guidelines in force at the year of pregnancy [11].

### 2.1. Design and Study Population

The maternity department of Saint-Pierre hospital is a tertiary care center that performs about 4300 deliveries per year, thus representing about 80% of deliveries of the Southern area of Reunion Island, but is the only level 3 maternity (the other maternity is a private level 1 hospital, which is not allowed to follow/deliver preeclamptic pregnancies). Reunion Island is a French overseas region in the Southern Indian Ocean. Virtually the whole population has then access to health care. This is provided free of charge by the French healthcare system, which combines freedom of medical practice with nationwide social security.

### 2.2. Definition of Exposure and Outcomes

From 2001 to 2020, we are sure that our tertiary unit covered all multiple pregnancies delivered in the southern part of the island.

SGA defined as gestational age-adjusted birth weight < 10th centile according to normal tables for our specific population (both sexes together).

Renal diseases were defined as patients with known pre-existing nephropathies without hypertension (glomerulopathies, tubulopathies, renal failure, diabetic nephropathies). Urological pathologies were excluded. Thyroid diseases were defined as hypo/hyperthyreoidism, goiter, thyroiditis, and prior thyroidectomy. Thrombophilias were defined as antiphospholipid syndrome, protein C or protein S deficiency, Factor 5 Leyden or other coagulation factors deficits at any time they were reported in the records (these were not systematically screened in all women as in a case-control study).

Preeclampsia was defined according to the World Health Organization recommendations [11] and the International Society for the study of Hypertension in Pregnancy [12] as the new onset of hypertension (BP ≥ 140 mmHg systolic or ≥90 mm Hg diastolic) at or after 20 weeks' gestation and substantial proteinuria (>0.3 g/24 h). Early onset preeclampsia (EOP) was defined as preeclampsia that resulted in the birth before 34 week's gestation. LOP was defined as preeclampsia associated with birth >34 week's gestation.

The "primipaternity" item (changing father for the index pregnancy) has been added in the database in 2018 and has been prospectively recorded since then. It is the sum of all primigravidas (and not primiparas) plus multiparous with changed partner for the index pregnancy. For the other years (2001–2017), we retrospectively looked at all free commentaries (possible in each record) for "changing father, changing paternity, new father, new partner etc." in multiparas (therefore probably non-exhaustive), but we retrieved 780 cases.

Screening of GDM is systematically made in all pregnant women in the second trimester: until 2016 it was the O'Sullivan test (50 g glucose, blood glucose level after 1 h). The threshold for hyperglycemia being 1.4 g/L. Since 2016, this test has been replaced in all women by a fasting glycemia in the first trimester, the threshold for positivity being 0.92 g/L, and the glucose tolerance test (GTT) at 24–28 weeks in all pregnant women. Those who have no GTT are only those who have a 1st trimester blood glucose over 1.26 g/L, which is considered to demonstrate Type 2 diabetes.

*2.3. Statistical Analyses*

Data are presented as numbers and proportions (%) for categorical variables and as mean and standard deviation (SD) for continuous ones. Comparisons between groups were performed using $\chi^2$-test; odds ratio (OR) with 95% confidence interval (CI) was also calculated. A paired t-test was used for parametric and the Mann–Whitney *U* test for non-parametric continuous variables. *p*-values < 0.05 were considered statistically significant. Epidemiological data were recorded and analyzed with the software EPI-INFO 7.1.5 (2008, CDC Atlanta, EPIDATA 3.0 and EPIDATA Analysis V2.2.2.183 (EpiData Association, Odense, Denmark, 2010. http://www.epidata.dk).

Furthermore, to validate the independent association of consensual risk factors on EOP or LOP, we realized a multiple regression logistic model. Variables associated with in bivariate analysis, with a *p*-value below 0.1 or known to be associated with the outcome in the literature were included in the model. A stepwise backward strategy was then applied to obtain the final model. The goodness of fit was assessed using the Hosmer-Lemeshow test. A *p*-value below 0.05 was considered significant. All analyses were performed using MedCalc software (version 12.3.0; MedCalc Software's, Ostend, Belgium).

*2.4. Ethical Approval*

This study was conducted in accordance with French legislation. As per new French law applicable to trials involving human subjects (Jardé Act), a specific approval of an ethics committee (comité de protection des personnes) is not required for this non-interventional study based on retrospective, anonymized data of authorized collections and written patient consent is not needed. Nevertheless, the study was registered on UMIN Clinical Trials Registry (identification number is UMIN000037012).

## 3. Results

During the 20-year period, there were 106,580 births in the south of Reunion Island of which 83,497 (78.4%, 81,834 pregnancies) in the University maternity of Saint-Pierre (the other births occurred in a single private clinic, level 1, not allowed to manage preeclamptic pregnancies and their births as well as multiple pregnancies).

There were 80,187 singleton newborns, 3164 twin newborns (1582 twin pregnancies), 138 triplet newborns (46 pregnancies) and 8 quadruplets (2 pregnancies).

During the period, we had 2120 preeclampsia (incidence 2.6% in the University maternity), of which 2001 preeclamptic singleton pregnancies and 119 twin pregnancies (incidence 7.5% in twin pregnancies).

Table 1 gives an overview of our singleton population, all preeclamptics vs. normotensive. PE mothers with were older than controls (28.9 vs. 27.8, *p* < 0.0001). Adolescents had less PE than controls (OR 0.73, *p* = 0.01), while women of 35+ years had a higher risk (OR1.55, *p* < 0.0001). PE women had a higher pre-pregnancy BMI (+3.1 kg/m$^2$) and were significantly overweight or obese (respectively OR 1.92 and 2.1, *p* < 0.0001). PE women were more prone to gestational diabetes (OR 1.45, *p* < 0.001), but not to pre-gestational diabetes (OR 0.89, *p* = 0.28). Concerning pre-existing diseases PE women had very significantly more chronic hypertension (OR 8.3, *p* < 0.0001), pre-existing renal or thyroid diseases respectively OR 5.4 (<0.0001) and 1.56 (*p* = 0.004). Compared to normotensive patients, women had a tendency to have more IVF and ovulation induction pregnancies respectively OR 1.56, *p* = 0.03 and OR1.7, *p* = 0.07). PE multiparas had much higher history of previous preeclampsia (OR 6.8, *p* < 0.0001) perinatal deaths (mainly intra-uterine fetal deaths OR 2.05, *p* < 0.0001).

**Table 1.** Maternal and pregnancy characteristics.

| | Preeclamptic Singleton Pregnancies N = 2001 | Controls Singleton Pregnancies N = 80,187 (%) | OR [95% CI] (%) | *p*-Values |
|---|---|---|---|---|
| Maternal age (years: mean ± sd) | 28.9 ± 7.0 | 27.8 ± 6.6 | | <0.0001 |
| adolescents | 61 (3.0) | 3277 (4.1) | 0.73 [0.57–0.91] | 0.01 |
| Age ≥ 35 years | 495 (24.7) | 14,198 (17.7) | 1.55 [1.39–1.7] | <0.0001 |
| Primiparity | 952 (47.6) | 29,920 (37.3) | 1.54 [1.4–1.7] | <0.0001 |
| Grand Multiparous women (≥5) | 199 (9.9) | 6532 (8.2) | 1.25 [1.08–1.46] | 0.001 |
| Marital status: Single | 756 (37.8) | 29,343 (36.7) | 1.05 | 0.31 |
| Years school ≥ 10 | 1086 (57.5) | 45,364 (58.8) | 0.95 | 0.23 |
| Pre-pregnancy BMI (mean ± sd) | 27.2 ± 6.9 | 24.8 ± 6.0 | | <0.0001 |
| ppBMI ≥ 25 kg/m$^2$ | 1014 (55.2) | 30,399 (39.5) | 1.92 [1.75–2.1] | <0.0001 |
| ppBMI ≥ 30 kg/m$^2$ | 567 (30.8) | 13,814 (17.9) | 2.1 [1.9–2.3] | <0.0001 |
| Gestational diabetes | 320 (16.0) | 9283 (11.7) | 1.45 [1.3–1.6] | <0.0001 |
| Pre-existing diabetes | 28 (1.4) | 1256 (1.8) | 0.89 | 0.28 |
| Chronic hypertension | 209 (10.5) | 1290 (1.6) | 8.3 [7.1–9.7] | <0.0001 |
| Renal diseases | 26 (1.3) | 217 (0.3) | 5.4 [3.6–8.1] | <0.0001 |
| Thyroid diseases # | 42 (2.1) | 1100 (1.4) | 1.56 [1.14–2.1] | 0.004 |
| In vitro Fertilization | 21 (1.0) | 547 (0.7) | 1.56 [1.0–2.4] | 0.03 |
| Ovulation induction | 9 (0.4) | 211 (0.3) | 1.7 [0.9–3.4] | 0.07 |
| History preeclampsia (multiparas) | 152/1290 (11.8) | 663/55,545 (1.2) | 6.8 [5.7–8.2] | <0.0001 |
| History of perinatal deaths (multiparas) | 112/1290 (8.7) | 2576/ 55,545 (4.4) | 2.05 [1.7–2.5] | *p* < 0.0001 |

# goiter, hypo-hyperthyroidy, thyroidectomy, thyroid node, thyroiditis.

Table 2 shows the decreasing rankings of CRUDE odds ratios of risk factors for all preeclampsia and for EOP. For all preeclampsia the ranking was: (1) history of preeclampsia in multiparas, (2) chronic hypertension, (3) renal diseases (4) MZ and DZ twins, etc.

For EOP, the ranking became: (1) MZ twin pregnancies, (2) ovulation induction pregnancies, (3) history of previous perinatal deaths in multiparas, and (4) chronic hypertension and previous history of preeclampsia (gestational diabetes being protective toward EOP, Or 0.69, *p* = 0.004).

In Table 3 we consider the logistic model with the outcome EOP some major risk factors remained as compared with crude odds ratios; (1) history of preeclampsia (1.6% of our multiparas), aOR 11.6, (2) chronic hypertension (1.6% of all our population) aOR 7.4, *p* < 0.0001, (3) pregnancies after ovulation induction aOR 4.8, *p* = 0.03. (4) Quite similar are MZ twin pregnancies (0.85% of all our pregnancies) 3.98, and DZ twin pregnancies (1.5% of our pregnancies), aOR 3.67, both *p* < 0.001., (5) first pregnancies (28% of our pregnancies: primigravid and multiparas with new father), aOR 3.0, *p* < 0.0001. In contrast, gestational diabetes (aOR 0.80, 12% of our pregnancies) was not a significant risk of EOP.

Specific to the EOP risk (not significant in LOP): monozygotic twins and medically induced pregnancies.

It is of note that both maternal ages (by increment of 5 years of age) and maternal pre-pregnancy BMI (by increment of 5 kg/m$^2$) had a slight constant increase (both aOR = 1.03, *p* = 0.004) on the EOP risk i.e., an increased risk of 2.7% by increment of 5 years of age or 5 kg/m$^2$ BMI (both coefficient 0.027, coefficients not shown in the tables).

**Table 2.** Crude Odds ratios. On the left, ranking of preeclampsia risk (as compared with controls: singleton pregnancies without hypertension) from the highest risk to the lowest. On the right, ranking of EOP risk. From the highest risk to the lowest (Reference/control: percentage of EOP in singleton preeclamptics: 32.0%).

| Risk Factors Ranking for Preeclamsia Risk | Incidence Preeclampsia (%) | Odds Ratios [95% CI] | p-Values | Risk Factors Ranking for EOP Risk | Percentage of EOP (%) | Odds Ratios [95% CI] | p-Values |
|---|---|---|---|---|---|---|---|
| previous preeclampsia (multiparas) | 152/811 (18.7) | 9.0 [7.5–10.9] | <0.0001 | Monochorionic twins | 17/26 (65.4) | 4.0 [1.8–10.2] | <0.001 |
| Chronic hypertension | 209/1289 (16.2) | 7.55 [6.5–8.8] | <0.0001 | ovulation induction | 9/16 (56.3) | 2.7 [1.01–7.3] | 0.02 |
| Renal diseases | 26/217 (12.0) | 5.3 [3.5–8.0] | <0.0001 | previous perinatal deaths (multiparas) | 57/116 (49.1) | 2.05 [1.4–3.0] | <0.001 |
| Monochorionictwins | 26/350 (7.4) | 3.1 [2.0–4.6] | <0.0001 | previous preeclampsia (multiparas) | 67/155 (43.2) | 1.6 [1.15–2.2] | 0.002 |
| Dichorionic twins | 84/1232 (6.8) | 2.85 [2.3–3.6] | <0.0001 | Chronic hypertension | 85/209 (40.6) | 1.45 [1.08–1.9] | 0.005 |
| Pre-existing diabetes | 80/1255 (6.4%) | 2.65 [2.1–3.3] | <0.0001 | Gestational diabetes | 79/320 (24.6) | 0.69 [0.53–0.90] | 0.004 |
| Ovulation induction | 16/301 (5.3) | 2.19 [1.3–3.6] | 0.002 | All singleton preeclamptics | 641/2001 (32.0) | Reference | - |
| IV Fertilization | 42/746 (5.6) | 2.3 [1.7–3.2] | <0.0001 | Thrombophilias | 7/14 (50.0) | 2.1 [0.74–6.1] | 0.12* |
| Thrombophilias | 14/278 (5.0) | 2.06 [1.2–3.5] | 0.006 | Renal diseases | 11/26 (42.3) | 1.55 [0.69–3.4] | 0.13 |
| previous perinatal deaths (multiparas) | 116/2655 (4.4) | 1.78 [1.47–2.1] | <0.0001 | Thyroid diseases | 16/42 38.1% | 1.3 [0.69–2.4] | 0.20 |
| Thyroid diseases | 42/1100 (3.8) | 1.55 [1.1–2.1] | 0.003 | Smoking | 62/180 (34.4) | 1.1 [0.80–1.5] | 0.25 |
| Gestational diabetes | 320/9283 (3.4) | 1.4 [1.2–1.6] | <0.0001 | Dichorionic twins | 26/84 (30.9) | 0.95 | 0.41 |
| Primipaternity | 747/22,662 (3.3) | 1.33 [1.2–1.4] | <0.0001 | IV Fertilization | 13/42 (30.9) | 0.95 | 0.44 |
| primigravidity | 691/21,898 (3.2) | 1.27 [1.16–1.38] | <0.0001 | Primipaternity | 224/747 (30.0) | 0.91 | 0.15 |
| All singleton pregnancies | 2001/80,187 (2.5) | Reference | - | primigravidity | 202/691 (29.3) | 0.88 | 0.09 |
| Smoking | 180/9923 (1.8) | 0.72 [0.62–0.84] | <0.0001 | Pre-existing diabetes | 23/80 (28.7) | 0.86 | 0.31 |

In Table 4, a logistic model with LOP as outcome is shown. Like for EOP (aOR 1.03 for maternal ages as well as for BMI), we still retrieve the constant increase of maternal ages and pre-pregnancy BMI but much stronger than for EOP: aOR = 1.04 for maternal ages (this time an increased risk of 4% by increment of 5 years of age, coefficient 0.042) and 1.05 for pre-pregnancy BMI (and an increased risk of 5% by increment of 5 kg/m$^2$ BMI, coefficient 0.051), both $p < 0.0001$.

**Table 3.** Adjusted Odds ratios. Outcome: EARLY ONSET preeclampsia EOP. Three logistic models including (1) singletons (2) monochorionic (3) dichorionic twins. Preeclamptic women PE (N = 2001), EOP (N = 641), LOP (N = 1360) versus controls, normotensive women (N = 78,085).

| | Singletons N = 641 EOP aOR | *p*-Value | MC N = 17/350 aOR | *p*-Value | DC N = 26/1232 aOR | *p*-Value |
|---|---|---|---|---|---|---|
| Ovulation induction | 5.6 [1.7–18.3] | 0.004 | 4.8 [1.1–20.3] | 0.03 | 3.1 [0.73–13.1] | 0.13 |
| previous perinatal deaths | 2.2 [1.5–3.1] | <0.0001 | 2.1 [1.45–3.1] | 0.0001 | 2.1 [1.5–3.04] | <0.0001 |
| previous Preeclampsia | 11.7 [8.5–16.0] | <0.0001 | 12.3 [8.4–17.0] | <0.0001 | 11.6 [8.4–16.0] | <0.0001 |
| Chronic HBP | 7.3 [4.1–6.3] | <0.0001 | 7.2 [5.1–10.1] | <0.0001 | 7.4 [5.3–10.3] | <0.0001 |
| IVF | 2.8 [1.01–7.7] | 0.05 | 2.1 | 0.30 | 1.2 | 0.72 |
| Primipaternity # | 3.0 [1.7–5.3] | 0.0001 | 2.96 [1.6–5.3] | 0.0002 | 3.02 [1.7–5.4] | 0.0002 |
| Maternal Age (increment/5 years) | 1.03 [1.01–1.05] | 0.005 | 1.03 [1.01–1.05] | 0.001 | 1.03 [1.01–1.05] | 0.004 |
| BMI (increment/5 kg/m$^2$) | 1.03 [1.01–1.04] | 0.002 | 1.03 [1.01–1.05] | 0.0006 | 1.03 [1.01–1.05] | 0.001 |
| Preexisting diabetes | 0.82 | 0.55 | 0.85 | 0.64 | 0.85 | 0.63 |
| Gest diabetes | 0.81 | 0.20 | 0.72 [0.52–1.0] | 0.06 | 0.81 | 0.20 |
| Renal diseases | 1.91 [0.66–5.5] | 0.23 | 2.0 [0.7–5.8] | 0.20 | 2.02 [0.70–5.8] | 0.19 |
| Thrombophilias | 1.66 [0.5–5.5] | 0.40 | 1.9 [0.55–6.1] | 0.31 | 1.8 | 0.33 |
| smoking | 0.90 | 0.56 | 0.94 | 0.73 | 0.87 [0.6–0.9] | 0.45 |
| previous abortions | 0.96 | 0.80 | 1.01 | 0.93 | 0.98 | 0.87 |
| MZ twins | - | - | 3.98 [1.6–9.9] | 0.003 | - | - |
| DZ twins | - | - | - | - | 3.67 [2.1–6.4] | <0.0001 |

# Multiparas with a new father.

In the LOP risk, the adjusted ranking became different than for the EOP risk:

First, similarities between EOP and LOP: (1) history of previous preeclampsia but much less than for EOP aOR 7.85 (vs. 11.6, EOP). (2) Chronic hypertension but again less than for EOP aOR 3.9, *p* < 0.0001 (vs. 7.4, EOP). (3) primipaternity aOR 3.5, *p* < 0.0001. (4) Dizygotic twin pregnancies, but less than for EOP aOR 2.1 vs. 2.7. (5) History of previous fetal deaths aOR 1.48, *p* = 0.007 (but less than for EOP aOR 2.1).

Disappearance of some risk factors in LOP as compared with EOP: MZ twin pregnancies and ovulation induction.

Specific to LOP (and not significant in EOP): renal diseases (glomerulo-tubulopathies, 0.3% of our pregnancies) aOR 2.9, *p* = 0.006, and diabetes (pre-existing diabetes aOR 1.6, *p* = 0.01 and GDM aOR 1.2, *p* = 0.03).

**Table 4.** Adjusted Odds ratios. Outcome: LATE ONSET preeclampsia LOP. Three logistic models including (1) singletons (2) monochorionic (3) dichorionic twins. Preeclamptic women PE (N = 2001), EOP (N = 641), LOP (N = 1360) versus controls, normotensive women (N = 78,085).

| | Singletons 1360/80,081 LOP aOR | *p*-Value | MZ Twins N = 9/350 aOR | *p*-Value | DZ Twins N = 57/1232 aOR | *p*-Value |
|---|---|---|---|---|---|---|
| Ovulation induction | 1.5 [0.37–6.3] | 0.56 | 1.35 [0.2–9.9] | 0.76 | 1.48 [0.4–6.2] | 0.58 |
| previous perinatal deaths | 1.48 [1.1–1.98] | 0.007 | 1.49 [1.07–2.07] | 0.02 | 1.48 [1.1–2.0] | 0.007 |
| previous Preeclampsia | 7.8 [6.1–10.1] | <0.0001 | 9.1 [6.8–12.0] | <0.0001 | 7.85 [6.1–10.2] | <0.0001 |
| Chronic HBP | 3.9 [3.0–5.1] | <0.0001 | 3.7 [2.7–5.1] | <0.0001 | 3.9 [3.0–5.1] | <0.0001 |
| IVF | 0.86 | 0.80 | 1.4 | 0.67 | 0.67 | 0.51 |
| Primipaternity # | 3.6 [2.5–5.2] | 0.0001 | 3.5 [1.6–5.3] | <0.0001 | 3.5 [2.4–5.1] | <0.0001 |
| Maternal Age (increment/5 years) | 1.04 [1.03–1.06] | <0.0001 | 1.04 [1.03–1.06] | <0.0001 | 1.04 [1.03–1.06] | <0.0001 |
| BMI (increment/5 kg/m$^2$) | 1.05 [1.04–1.06] | <0.0001 | 1.05 [1.04–1.07] | <0.0001 | 1.05 [1.04–1.06] | <0.0001 |
| Preexisting diabetes | 1.57 [1.09–2.3] | 0.01 | 1.6 [1.1–2.4] | 0.008 | 1.6 [1.1–2.3] | 0.01 |
| Gest diabetes | 1.2 [1.01–1.5] | 0.04 | 1.2 [1.01–1.5] | 0.04 | 1.2 [1.01–1.5] | 0.03 |
| Renal diseases | 2.9 [1.3–6.2] | 0.007 | 2.6 [1.07–6.1] | 0.03 | 2.9 [1.3–6.3] | 0.006 |
| Thrombophilias | 2.0 [0.9–4.7] | 0.10 | 1.5 [0.55–4.3] | 0.40 | 2.0 [0.86–4.7] | 0.10 |
| smoking | 0.80 [0.62–1.03] | 0.09 | 0.70 [0.52–0.95] | 0.02 | 0.80 [0.6–1.04] | 0.09 |
| previous abortions | 1.02 | 0.83 | 1.1 | 0.33 | 1.1 | 0.87 |
| MC twins | - | - | 1.44 [0.45–4.6] | 0.53 | - | - |
| DC twins | - | - | - | - | 2.07 [1.3–3.4] | 0.003 |

# Multiparas with a new father.

Figure 1 summarizes a visualization of all adjusted odds ratios calculated in Tables 3 and 4: independent risk factors for EOP and LOP controlling for the two major confounders: maternal ages (both risks for EOP and LOP) and maternal pre-pregnancy BMI (major risk factor for LOP).

(1) Specific risk of EOP: monochorionic twins (and in our experience especially female pairs) and pregnancies after ovulation induction. (2) Common risk factors between EOP and LOP but with a much higher risk of EOP: chronic hypertension and history of preeclampsia. (3) Common risk factors between EOP and LOP but with a similar risk: dichorionic twins, new paternity, history of perinatal deaths. (4) Specific risk of LOP (moreover the known effect of overweight and obesity): renal diseases and gestational diabetes mellitus.

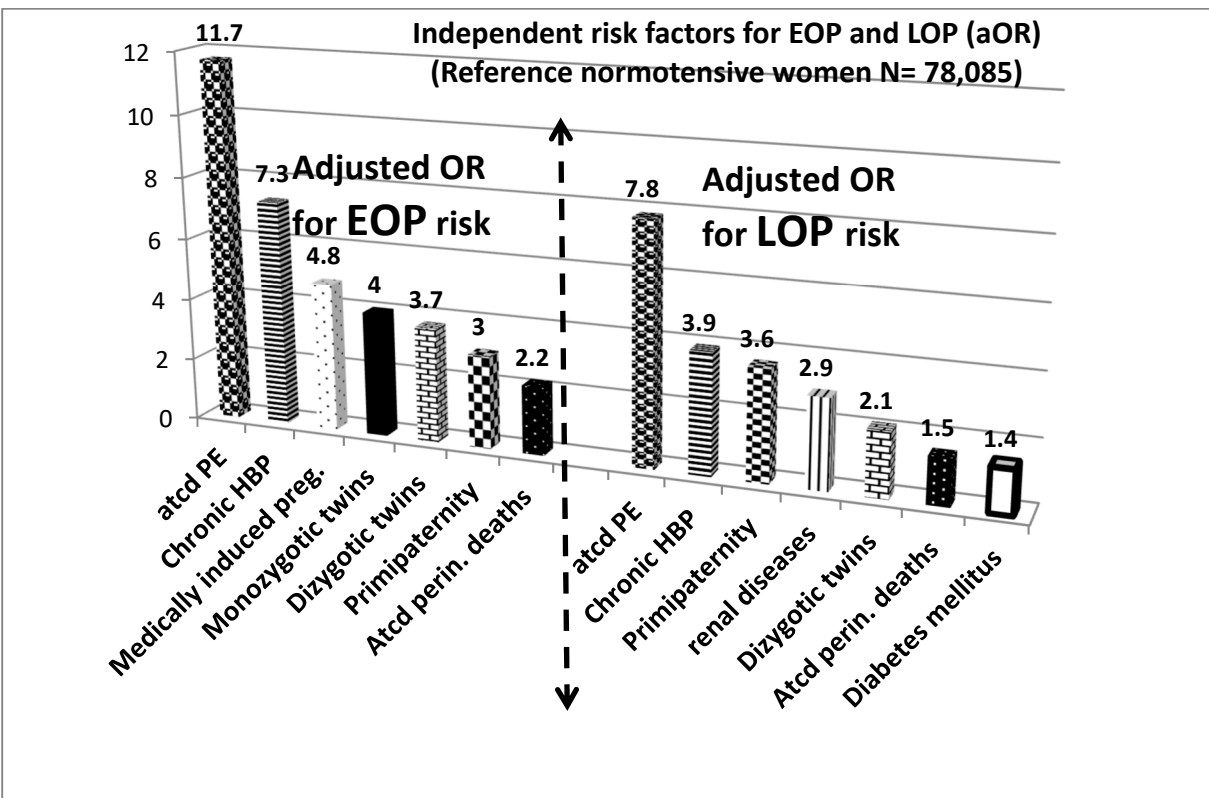

**Figure 1.** Independent risk factors for EOP and LOP (adjusted odds ratios) controlling for the two major confounders: maternal ages (both risks for EOP and LOP) and maternal pre-pregnancy BMI (major risk factor for LOP). Increment by 5 years of age and by 5 kg/m² for and maternal pre-pregnancy BMI.

## 4. Discussion

Our study confirms that the major risk factors for EOP are chronic hypertension, history of preeclampsia, ovulation induction, history of perinatal deaths and twin pregnancies. Reading the international literature, it seems that an international consensus is comprised of 11 risk factors: multiple pregnancies, chronic hypertension, preeclampsia or hypertension in a previous pregnancy, diabetes, nulliparity, renal disease, "Thrombophilias"-anti phospholipid syndrome, maternal age > 40 years, body mass index > 35 kg/m², family history of preeclampsia, assisted reproduction [12–16]. Especially for Lisonkova and Joseph, where "African-American race, chronic hypertension, and congenital anomalies were more strongly associated with early onset preeclampsia, whereas younger maternal age, nulliparity, and diabetes mellitus were more strongly associated with late onset disease." [13]. This study wishes to contribute to epidemiological studies on risk factors for preeclampsia, especially focusing on EOP.

### 4.1. Specificities of Twin Pregnancies

Several studies examined the rate of PE in twins in relation to chorionicity in the 1970/1990's, cited by Savvidou et al. in 2001 [17], giving conflicting results but determination of zygocity was imperfect in this period. Determination of chorionicity could be performed prenatally accurately and non-invasively by ultra sonography during the first trimester of pregnancy (10–14 weeks of gestation) by the lambda sign described in 1996 [18]. Since then seven studies examined the rate of PE in twins in relation to chorionicity: in six studies the rate was similar in dichorionic and monochorionic twins [16,17,19–22], but in one study the rate was twice as high in dichorionic than in monochorionic twins [23]. In our experience, we confirm that twin pregnancies globally have a relative risk of multiplicated

by 3 to experience preeclampsia as compared with singletons [9], but higher specific high risk of the dangerous early form EOP in mono chorionic twins [9].

### 4.2. Prevention of EOP

Regarding prevention of EOP, there is now good evidence that 150 mg of aspirin < 16th week of gestation will halve the risk [24,25] except in patients with chronic hypertension, where no beneficial effect is evident [25]. This prevention may also affect the rate of the iatrogenic preterm birth by delaying the medical decision to induce the newborns births/perform a cesarean section [26]. Twin pregnancies are certainly highly involved in these indications [27,28].

Late onset preeclampsia LOP, 34th week of gestation onward (90% of preeclampsia cases in developed 'Western lifestyle countries'). The particular case of gestational diabetes (GDM) and optimal gestational weight gain (GWG) surveillance in overweight/obese women.

Pre-existing diabetes is a well-established risk of diabetes, and several studies previously proposed GDM as a risk factor (risk X by 2 or 4 [29]). Reports associate diabetes and preeclampsia [13,30,31], especially with LOP [13]. However, a few studies [32,33] do not find such an association when controlling for maternal weight and age, as in a very recent study, in a cohort of 15,000 pregnant women in Beijing, China [34]. In our prospective cohort also, after controlling for maternal pre-gestational BMI and maternal ages, gestational diabetes is no longer an independent risk of preeclampsia, neither for EOP (even protective, Table 3) and poorly for late onset PE (aOR 1.2, $p = 0.04$, Table 4). This confirms what we previously described [5]. These findings suggest also that a certain degree of mild hyperglycaemia per se does not adversely affect maternal-placental homeostasis [5].

For the LOP risk, our multivariate model identifies some risk factors common to the EOP risk (Tables 3 and 4, Figure 1): History of preeclampsia, chronic hypertension, history of perinatal death. Renal diseases and gestational diabetes mellitus (small association OR 1.2, $p = 0.03$) seem to be specific to the LOP risk. However, here, we would like to propose a 'new' promising approach: recently two different teams from two different parts of the world (USA and Reunion Island, Indian ocean) [5,35] described that late onset preeclampsia (and much less early onset preeclampsia) is largely and specifically linearly associated with higher maternal Body Mass Index (BMI). Further research is urgently required in order to properly understand the main drivers and pathways on how the cardiometabolic syndrome leads to late onset preeclampsia [36]. Importantly, we recently demonstrated that having a high BMI does not automatically translate to a higher risk of term preeclampsia. Obese patients can significantly decrease their risks by fine tuning of their gestational weight gain (GWG) [8,37–40].

### 4.3. Strengths

The strengths of this study are mostly related to the homogeneity of data in such a large cohort as they were collected in a single center (no intercenter variability) and not based on national birth registers but directly from medical records (avoiding inadequate codes).

The Centre Hospitalier Universitaire Sud Reunion's maternity ward (Level 3, European standards of care) is the only public hospital in the southern part of Reunion Island (Indian Ocean, French overseas department). It serves the whole population of the area (approx. 360,000 inhabitants, and 5100 births per year). With 4300 births per year, the university maternity ward represents 82% of all births in the south of the island. However, as a level 3 (the other maternity ward is a private clinic, level 1), we are sure all the preeclampsia cases as well as all multiple pregnancies were referred to our hospital during the 20 year period. This is therefore a real population-based study.

### 4.4. Weakness

As a limitation of the study, we have to consider the retrospective nature of the study that although the amount of information that is recorded is comprehensive, some characteristics may be missing, such as length of sexual relationship and/or primipaternity.

The presence of thrombophilia was not systematically screened in all women (cases and references). However, every time a woman was known to have one of these characteristics, they were scrupulously included in the database. Additionally the item on partner change was only added to the database in 2018. For the other years (2001–2017), we retrospectively looked at all free commentaries (possible in each record) for "changing father, changing paternity, new father, new partner etc.". We did not test the interval between pregnancies nor possible aspirin use.

## 5. Conclusions

This study confirms that maternal ages over 35 years, chronic hypertension, history of preeclampsia, ovulation induction, history of perinatal deaths, and possibly twin pregnancies (in our experience, especially mono chorionic twin pregnancies) are significant risk factors for EOP, the fearsome early onset form of preeclampsia. It is of note that in both multivariate models for EOP and LOP, we noticed as an independent risk factor: "primipaternity" (i.e., the combination of primigravid and multiparas with a new male partner yes/no, without any notion of duration of cohabitation before conception). This approach of paternity may urgently require adequately powered prospective trials as we hypothesize that it should be new couples with less than six months of sexual cohabitation before conception which should be those especially at risk of EOP [40] (and beneficiaries of aspirin prophylaxis). Therefore, clinicians should add as a systematic question the inquiry of paternity in all beginning of prenatal visit in any pregnancy and also evaluate the length of sexual cohabitation before conception in the case of a new father.

**Author Contributions:** P.-Y.R. participated at all the stages of the study (data collection, analysis, writings etc.). G.D. verified all the epidemiological calculations and participated deeply to the data analysis. G.D., M.S. and S.I. expertised the analysis, the text and the final writings (and the English Language). S.I., M.B., F.B. and B.B. participated at the data collection, analysis and writings. All authors have read and agreed to the published version of the manuscript.

**Funding:** This research received no external funding besides the normal existence of the South-Reunion perinatal database.

**Institutional Review Board Statement:** Not applicable.

**Informed Consent Statement:** Not applicable.

**Data Availability Statement:** Data accessible with robillard.reunion@wanadoo.fr.

**Conflicts of Interest:** The authors declare no conflict of interest.

## Abbreviations

| | |
|---|---|
| EOP | early onset preeclampsia |
| LOP | late onset preeclampsia |
| ppBMI | pre-pregnancy maternal body mass index |
| GDM | gestational diabetes mellitus |
| CH | chronic hypertension |
| HP | history of preeclampsia in a preceding pregnancy |
| FGR | fetal growth restriction |
| SGA | small for gestational age |
| IVF | in vitro fertilization |
| MC | monochorionic twins |
| DC | dichorionic twins |

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
