# Peer review of "Risk Factors for Early and Late Onset Preeclampsia in Reunion Island: Multivariate Analysis of Singleton and Twin Pregnancies. A 20-Year Population-Based Cohort of 2120 Preeclampsia Cases"

_2673-3897, doi:10.3390/reprodmed2030014_

Round 1

Reviewer 1 Report

The manuscript focuses on assessing the risk of preeclampsia in twin (mono- and dizygotic) pregnancies compared to single pregnancies. This is a retrospective study covering 20 years of observation.

Unfortunately, there are some limitations:

The authors should present their previous results in a narrower sense in “Introduction”. “Discussion” seems too limited. Please discuss your previous observations in this section. Please find studies of another authors concentrating on the risk of preeclampsia in multiple gestation.

The authors informed in “Results” that there were 83,555 births of 81,834 pregnancies in their hospital within 20 years. There is also a piece of information that there were 80,187 singleton pregnancies, 1,232 twin dizygotic pregnancies and 350 twin monozygotic pregnancies, i.e. 81,769 in total (Table 1). Please add details of triplets, quadruplets, etc. during this period.

Please re-convert the sequence of variables in Table 1. For example: items 8. (“Adolescents”) and 10. (“Age more than 35 years”) should be closer to item 1. (“Maternal age (years mean +/- SD)”) (i.e. as items 2. and 3.). Similarly, items 7. (“Primiparity) and 9. (“Grand multiparous”) are related to items 2. (“Gravidity”) and 3. (“Parity”).

The authors mentioned gestational weight gain (GWG) in “Discussion”. Is it possible to add this parameter to statistical analysis? GWG is an important risk factor for preeclampsia.

What does "Atcd" mean? Please find “Atcd” in the tables.

Please change the order of analyzed parameters on the right side (EOP risk) in Table 2 – according to the Odds Ratios.

How to explain the fact that a diagnosis of gestational diabetes mellitus during pregnancy is a protective factor for EOP? Please discuss this interesting result in “Discussion”.

The authors emphasized that one of the main risk factors for EOP is monozygotic twin pregnancy – “in their experience”. Please check the correctness of p-values in Table 3 (MZ twins with p= 0.003 vs. DZ twins with p <0.0001).

Author Response

Modifications REVIEWER 1 in dark blue in the text. REVIEWER 2 in Red in the revised manuscript

Reviewer 2 Report

The authors examined the risk factors of early and late-onset preeclampsia in Reunion island. The results of this study are of some interest. However, there are several problems to resolve. The reviewer’s comments are listed below.

Introduction

The Introduction is too long and redundant. Please reduce to 3 paragraphs with about 500 words.

Materials and Methods

What is ap.? Please do not use abbreviations without definition.

Please cite previous studies regarding the definition of preeclampsia.

The traditional definition of pre-eclampsia, according to the International Society for the Study of Hypertension in Pregnancy (ISSHP), is a new onset of hypertension (blood pressure ≥140 mmHg systolic or ≥90 mmHg diastolic) at ≥20 weeks of gestation and proteinuria (≥300 mg/24 h or protein-to-creatinine ratio >30 mg/mmol or ≥2+ on dipstick testing). This has been revised recently by the ISSHP and American College of Obstetricians and Gynecologists to include cases without proteinuria, but with evidence of renal, hepatic, or hematological dysfunction. Did the definition of HDP remain consistent throughout the study period (2001–2020)?

Results

Table 1

It does not make sense to me why the authors showed the characteristics of pregnant women with twins. In the first table, I think the authors may show the characteristics of women with and without preeclampsia. (Non-preeclampsia versus preeclampsia).

I prefer to use in vitro fertilization instead of in vitro fecundation.

Table 2

Please revise the title of Table 2.

What is Atcd? Please define the abbreviations in the legend of Tables.

Please unify the size and type of font.

Table 3

Please revise the title of Table 3.

What does # indicate?

Please delete the unnecessary underline.

Table 4

Please revise the title of Table 4.

Limitations

Although the authors mentioned the effect of aspirin on the prevalence of preeclampsia, the authors did not show the rate of aspirin use. The authors may add this point as a limitation of this study.

Author Response

(The authors gave the same response as above.)

Round 2

Reviewer 1 Report

Dear Authors,

Thank you for attempting to address my concerns.

I accept the manuscript in the present form.

Reviewer 2 Report

The authors addressed could revise the manuscript according to my previous comments.